

# A joint probabilistic index for objective drought identification: the case study of Haiti

Beatrice Monteleone[1], Brunella Bonaccorso[2], and Mario Martina[1]

[1]Scuola Universitaria Superiore IUSS Pavia, Pavia, 27100, Italy
[2]Department of Engineering, University of Messina, S. Agata, Messina, 98166, Italy

**Correspondence:** Beatrice Monteleone (beatrice.monteleone@iusspavia.it)

**Abstract.** Since drought is a multifaceted phenomenon, more than one variable should be considered for a proper understanding of such extreme event in order to implement adequate risk mitigation strategies such as weather or agricultural indices insurance programs, or disaster risk financing tools. This paper proposes a new composite drought index that accounts for both meteorological and agricultural drought conditions, by combining in a probabilistic framework two consolidated drought

indices: the Standardized Precipitation Index (SPI) and the Vegetation Health Index (VHI). The new index, called Probabilistic Precipitation Vegetation Index (PPVI), is scalable, transferable all over the globe and can be updated in near-real time. Furthermore, it is a remote-sensing product, since precipitation are retrieved from satellite and the VHI is a remote-sensing index. In addition, a set of rules to objectively identify drought events is developed and implemented. Both the index and the set of rules have been applied to Haiti. The performance of PPVI has been evaluated by means of the Receiver Operating Characteristics curve and compared to the ones of SPI and VHI considered separately. The new index outperformed SPI and VHI both

in drought identification and characterization, thus revealing potential for an effective implementation within drought early warning systems.

## 1   Introduction

Droughts affect every year an increasing number of people. In the years from 2014 to 2018 more than 70 drought events have

been reported all over the world and about 450 million people suffered because of drought-related impacts (CRED, 2017). Due to its complexity, various definitions of the phenomenon have been proposed by different institutions, such as the World Meteorological Organizations (WMO), the Food and Agriculture Organization (FAO) and the United Nations Convention to Combat Desertification (UNCCD). All the institutions focus their attention on a specific aspect of drought: the WMO on the lack of precipitation, the FAO on the decline in crop productivity and the UNCCD on the loss of arable land.

In addition, the quantification of drought effects is a complicated task, since drought impacts are non-structural, widespread over large areas, and of different type and magnitude within the drought-affected area, also depending on economic, social and environmental system vulnerabilities (Wilhite, 2000).

Drought identification through an objective and automatic determination of drought onset, termination and severity allows the timely adoption of appropriate risk management strategies, such as weather index insurance programs (Barnett and Mahul,



2007), agricultural index insurance programs (Jensen and Barrett, 2017), disaster financing (Guimarães Nobre et al., 2019; Linnerooth-Bayer and Hochrainer-Stigler, 2015) and early action planning (Drechsler and Soe, 2016).

Drought features are usually determined through the use of two instruments: indicators, which are variables and parameters used to assess drought conditions (such as precipitation, temperature, and others), and indices, which are numerically computed values from meteorological or hydrological inputs (World Meteorological Organization and Global Water Partnership,

2016). More than 100 indices have been developed by the scientific community (Zargar et al., 2011), each one focusing on a specific aspect of drought (meteorological, hydrological, agricultural and so on). Meteorological drought is related to precipitation shortages; hydrological drought refers to periods of precipitation shortfall on surface or subsurface water supply **??**, while agricultural drought is conventionally linked to soil moisture deficit. Insufficient soil moisture leads to crop failure and consequent yield reduction; therefore the first economic sector suffering because of drought is agriculture, particularly in those

areas where it relies on rainfall. A deeper understanding of agricultural drought dynamics can promote the adoption of risk reduction strategies, such as crop insurance programs.

In recent years various remote-sensing indices have been developed and can be employed in agricultural drought monitoring. The most widespread is the Normalized Difference Vegetation Index (NDVI), which uses NOAA AVHRR satellite data to monitor vegetation greenness (Kogan, 1995a). The main advantages of the NDVI are the very high spatial resolution and the

global coverage. The NDVI has already been applied in drought monitoring, such as in (Gu et al., 2008). Many products were derived from the NDVI, such as the Vegetation Condition Index (VCI), which compares the current NDVI to the range of values observed in the same period in previous years (Liu and Kogan, 1996; Kogan, 1995b) and the Standardized Vegetation Index (SVI), which describes the probability of vegetation condition deviation from normal (Peters et al., 2002). A suite of agricultural drought indices is presented in Table 1.

Since drought is a complex phenomenon, a single index or indicator can be insufficient to fully characterize drought severity and extent. The combination of more than one indicator can be precious to evaluate all the variables involved in drought monitoring, such as precipitation, soil moisture, and streamflow. Over the past 20 years many composite indicators, relying on two or more drought indices or indicators, have been proposed to overcome the issues related to the use of a single variable. Table 2 shows a list of selected composite indices that can be used in agricultural drought monitoring since, in their formulation

soil moisture, vegetation condition or variables related to water availability for plants are included.

Multiple methods for taking into account the multivariate behaviour of drought have been explored (Hao and Singh, 2015; Hao, 2016). The VegDRI, for example, uses a data mining approach to combine multiple inputs such as the SPI, the NDVI and the Palmer Drought Severity Index (PDSI). A weighted linear combination of the inputs is quite common; it is applied to construct the Composite Drought Indicator (CDI) for Morocco, the Vegetation Health Index (VHI) and the Objective Blend of

Drought Indicators (OBDI). The United States Drought Monitor (USDM) also applies a weighted linear combination of the inputs but adds an expert judgment to define the drought class.

In the last years multiple studies focused the attention on modelling the joint behaviour of two drought characteristics or indices applying bivariate or multivariate statistical approaches. In various cases bivariate distributions are developed by means of copulas as in (Serinaldi et al., 2009) and (Bonaccorso et al., 2012), where the joint behaviour of various drought properties is





investigated; or in (Shiau, 2006), where two-dimensional copulas are employed to study the joint behaviour of drought duration
and severity in Taiwan. (Shiau et al., 2007) investigates also the hydrological droughts of the Yellow River in China using a
bivariate distribution to model drought duration and severity jointly. Trivariate Plackett copula is used in (Songbai and Singh,
2010) to model drought duration, severity and inter-arrival time jointly.

The use of copulas to quantify the joint behaviour of drought indices is gaining popularity too. Many drought indices derived
by multivariate distributions have been proposed. For example the Multivariate Standardized Drought Index (MSDI) (Hao and
Aghakouchak, 2013), which combines the SPI and the Standardized Soil Moisture Index (SSI), uses copula to form joint
probabilities of precipitation and soil moisture content, while the Joint Drought Index (JDI) (Kao and Govindaraju, 2010) does
the same for obtaining the joint probabilities while considering precipitation and streamflow. AMDI-SA combines two drought
indices, the Modified SPI, and the Modified SSI, employing both the copula concept and the Kendall function (Bateni et al.,
2018). The use of copulas seems promising and is highly effective when dealing with two or more variables. An advantage of
copula functions is the fact that the index derived from this approach has a probabilistic form.

Both single and composite indices for agricultural drought monitoring showed some limitations, highlighted in Table 1 and
Table 2. Single indices often rely on multiple inputs or are available only for some locations or identify all types of vegetation
stresses. In any cases single indices do not account for the multivariate nature of drought. Composite indices often rely on
relatively new datasets; in many cases a short period of record is available (for example the VegDri records start in 2009) or
the index is not available in near-real time; some of them are specifically designed for a well identified region (the OBDI and
the USDM are available only for the USA, the Combined Drought Indicator (CDI) only for Europe); other indices do not
consider the meteorological aspect of drought (Temperature Vegetation Index, TVX, and Vegetation Temeprature Condition
Index, VTCI, are based on the NDVI and the land surface temperature); other ones do not have a sufficiently refined spatial
resolution (MSDI). Most of them, with the exception of AMDI-SA and MSDI are not expressed in probabilistic terms, therefore
uncertainty quantification and evaluation is not an easy task.

In this paper, we propose:

1. A new drought index, the Probabilistic Precipitation Vegetation Index (PPVI), that takes the advantage of well consol-
idated indices, the Standardized Precipitation Index (SPI) (Mckee et al., 1993) and the Vegetation Health Index (VHI)
(Kogan, 1997) and tries to overcome their individual limitations by coupling them in a probabilistic framework through
the use of a bivariate normal distribution function.

2. A framework to identify a drought event using the new index, i.e. a set of rules for the definition of a drought event.
When the set of conditions is verified, a drought event is identified based on the new index. Otherwise, no drought event
is identified.

With respect to the indices already available in literature, we will show in this paper that the new index has some interesting
features:

• it is able to identify drought-driven events of vegetation stress;





- it is parsimonious in terms of number of inputs required;

- it is a remote sensing product with high spatial and temporal resolution;

- it is based on quasi-near real time datasets, with a relatively short latency time (less than one week);

- more than 30 years of records are available at global scale for its calibration.

The paper is structured as follows: Sect.2 describes the datasets employed in the development of the new index and presents the methodology used to combine the SPI and the VHI; Sect. 3 illustrates the application to the case study, shows the validation process of the new index and compares the performance of the new index to those of the SPI and the VHI considered 100 separately; in addition the advantages related to the adoption of the index and the possible applications in agricultural drought risk management are summarized.

## 2 Datasets and Methods

### 2.1 Datasets

Two remote-sensing datasets were used: one for precipitation and the other for the VHI. Precipitation was retrieved from the 105 satellite-only Climate Hazard Group Infrared Precipitation (CHIRP) dataset. CHIRP has a quasi-global coverage (50°S - 50°N), high spatial resolution (0,05°) and daily, pentadal and monthly temporal resolution. Records start from 1/1/1981. CHIRP was chosen because it has been specifically developed to monitor agricultural drought. The use of CHIRP instead of CHIRPS (the Climate Hazard Group Infrared Precipitation with Stations) is related to the data latency time, which is shorter in the case of CHIRP since it doesn't include data from weather stations. The development and the main characteristics of the dataset are 110 described in (Funk et al., 2015). In the present study CHIRP with a daily temporal resolution was used to have the possibility to compute weekly precipitation. Data are available on the project website (Climate Hazard Group, 1999).

The Vegetation Health Index was retrieved from the Global Vegetation Health Products (Global VHP) of the National Oceanic and Atmospheric Administration Center of Satellite Applications and Research (Kogan, 1997). Data can be retrieved at the NOAA website (NOAA, 2011). The dataset contains Blended-VHP derived from VIIRS (2013-present) and AVHRR 115 (1981-2012) GAC data. The dataset has 4km spatial resolution, weekly temporal resolution, and global coverage. Both the selected datasets are freely available.

### 2.2 Methods

#### 2.2.1 The Standardized Precipitation Index

As previously mentioned, two consolidate drought indices were combined: the SPI and the VHI. The SPI was selected because 120 it is a commonly used index to detect meteorological drought, it is standardized, therefore SPI values can be compared even in different climate regimes and it is recommended by the WMO (World Meteorological Organization, 2009).





SPI computation is based on a long-term precipitation record for a desired period. The precipitation record is then fitted to a probability distribution (in this work a gamma distribution was used), which is then transformed into a normal distribution. Traditionally monthly precipitation records are employed, and SPI is computed aggregating precipitation at a predefined

timestep (for example 1 month, 3 months, 6 months, 9 months and 12 months are the aggregation periods suggested by the WMO (World Meteorological Organization, 2009)).

In the present work, weekly precipitation records were used. The SPI aggregation period was then selected and the index, computed over one of the the traditional aggregation periods, was updated every week. SPI is normal distributed by definition. Conventionally drought starts when SPI is lower than -1 and ends when SPI comes back to the value of 0 (Mckee et al., 1993).

Drought classification according to SPI, as proposed in (Mckee et al., 1993), is reported in Table 3. The percentages reported in the third column of Table 3 indicate the probability for SPI values to fall within the range reported in the second column of the same table.

### 2.2.2   The Vegetation Health Index

The VHI is a remote-sensing index developed to include the effects of temperature on vegetation; in fact, it combines the

VCI with the Temperature Condition Index (TCI) (Kogan, 1995a), which is another remote-sensing index used to determine vegetation stress caused by temperature and excessive wetness. One drawback of the VHI is the impossibility to identify the cause of the vegetation stress; in fact, vegetation can suffer because of various events: excessive wetness, pests, fires, droughts or others. It is a biophysical indicator of a lack of precipitation but can also be seen as representing drought impacts on the ground (Bachmair et al., 2016). It goes from 0, which stands for vegetation in very bad conditions to 100, meaning perfectly

healthy vegetation. The classification scheme of VHI, as proposed in (Dalezios et al., 2017), is presented in Table 4.

The VHI is standardized according to the following equation:

$$VHI_{st} = \frac{VHI - \overline{VHI}}{\sigma} \tag{1}$$

where $\overline{VHI}$ is the mean of the distribution and $\sigma$ its standard deviation. The standardized variable, $VHI_{st}$, has a distribution with 0 mean and 1 as standard deviation.

### 2.2.3   The Probabilistic Precipitation Vegetation Index (PPVI)

The Probabilistic Precipitation Vegetation Index (PPVI) is a composite index that takes into account both meteorological drought through the SPI, and agricultural drought conditions by including the VHI.

In order to combine the two consolidated indices the following preparatory steps are performed:

1. Extraction of the area under study from both the datasets;

2. Regridding of both precipitation and the VHI to bring them to the same spatial resolution (0,05°);

3. Aggregation of precipitation at weekly timescale (CHIRP has daily temporal resolution);



4. Computation and weekly update of SPI according to the methodology proposed in (USDA Risk Management Agency et al., 2006), where precipitation are fitted to a gamma distribution. The goodness of fit to the gamma distribution has been verified by means of probability plot.

5. Standardization of the VHI, as previously described.

The combination of SPI and VHI is performed using a bivariate normal distribution function, as it is defined by (Kotz et al., 2000). The normality of the SPI and $VHI_{st}$ distributions has been verified as will be shown in Sect. 3.2. Therefore it is acceptable to assume that the joint probability of the two considered distributions takes the form of the bivariate normal for correlated variables:

$$160 \quad f(s,v) = \frac{1}{2\pi\sigma_s\sigma_v\sqrt{1-\rho^2}} \exp\left(-\frac{1}{2(1-\rho^2)}\left[\frac{(s-\mu_s)^2}{\sigma_s} + \frac{(v-\mu_v)^2}{\sigma_v} + \frac{2\rho(s-\mu_s)(v-\mu_v)}{\sigma_s\sigma_v}\right]\right) \quad (2)$$

where the following notation is adopted: the *SPI* is identified as *s* and the $VHI_{st}$ is identified as *v*. The mean and the standard deviation of the SPI distribution $f(s)$ are respectively, by construction, $\mu_s = 0$ and $\sigma_s = 1$ and the mean and standard deviation of the $VHI_{st}$ distribution, $f(v)$ are respectively $\mu_v = 0$ and $\sigma_v = 1$. The covariance matrix $\Sigma$ and the correlation coefficient $\rho$ are defined according to Eq. 3 and Eq.4 respectively, where $\sigma_{sv}$ is the covariance between *s* and *v*.

$$165 \quad \Sigma = \begin{bmatrix} \sigma_s^2 & \rho\sigma_s\sigma_v \\ \rho\sigma_s\sigma_v & \sigma_v^2 \end{bmatrix} \quad (3)$$

$$\rho = \frac{\sigma_{sv}}{\sigma_s\sigma_v} \quad (4)$$

To check the assumption of normality for the joint distribution, the joint probability values, retrieved from Eq.2 are plotted against the bivariate empirical cumulative distribution values (Fig.1), as done in (Kao and Govindaraju, 2010). The bivariate empirical copula for the random variables *s* and *v* has been evaluated according to (Nelsen, 2006) using the following equation:

$$170 \quad C\left(\frac{i}{m}, \frac{j}{m}\right) = \frac{\#(s \leq s_{(i)}, v \leq v_{(j)})}{m} = \frac{m_1}{m} \quad (5)$$

where $s_{(i)}$ and $v_{(j)}$, $(1 \leq i, j \leq m)$ are ordered statistics of the *SPI* sample of size $m$, $m_1$ is the number of samples $(s_{(k)}, v_{(k)})$ satisfying $(s_{(k)} \leq s_{(i)}$ and $v_{(k)} \leq v_{(j)})$ with $1 \leq k \leq m$. The resulting plot is shown in Fig.1.

Since the data lays on the 45°line it is fair to assume that the joint probability $f(s,v)$ is normal. Therefore, a normalization of the index is performed through normal quantile transformation.

By keeping the same probability intervals of the SPI, we can compute the PPVI values for the drought classification as it is shown in Table 5.

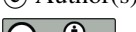



### 2.2.4 Identification of drought events

Once the index is defined the set of rules to establish when a grid cell is in drought should be identified. In particular, two parameters have to be identified:

1. A threshold $Z$ of the index that marks the beginning of a drought in a grid cell.

2. A threshold $z$ that marks the end of a drought in the same grid cell.

According to the model here proposed a drought in a grid cell starts when the index is lower than $Z$ and ends when the index is above to $z$. Then regional drought events are defined. Again, two parameters are identified: $N$ and $n$. A drought events starts if more than $N$ grid cells are in drought conditions and ends if less than $n$ grid cells are in drought conditions.

### 185  2.2.5 Skill assessment

Observations of drought are compared with the model outputs for various combinations of thresholds $Z$, $z$, $N$ and $n$. The Receiver Operating Characteristic (ROC) curve is used for this comparison. The ROC curve was at first used in signal detection; its use in meteorological applications is documented and well described in (Joliffe and Stephenson, 2012). The ROC curve is employed to classify instances, as in the present case. The ROC curve was already employed in various studies to compare the

performance of a model versus observations with varying thresholds (Zhu et al., 2016; Khadr, 2016). The contingency matrix (shown in Table 6) is a two by two matrix to visualize the disposition of a set of instances. True positive or hits are represented by the weeks that are reported to be in drought conditions in the observations and are correctly identified as drought weeks by the model. True negatives (correct rejections) are represented by those weeks that are not in drought according to both the observations and the model. Those weeks that are recorded as drought according to the observations but are not identified

as drought weeks by the model, are considered as false negatives (missing events), while false positives (false alarms) are represented by the weeks that are not in drought conditions according to the observations but are identified as drought weeks by the model. In this paper for each combination of thresholds $Z$, $z$, $N$ and $n$, Probability of Detection (POD), or hit rate, and Probability of False Detection (POFD), or false alarm rate, are computed according to (Joliffe and Stephenson, 2012) with the following equations:

$$POD = \frac{TP}{TP + FN} \tag{6}$$

$$POFD = \frac{TN}{TN + FP} \tag{7}$$

where $TP,TN,FP$ and $FN$ are defined as in Table 6.

The optimal threshold for a ROC curve is the one for which the distance from the 45°degree line is maximum (Zhu et al.,

2016). The performances of the model based on PPVI in identifying drought events have been evaluated on the case study described in the next section.





### 2.2.6  Case study

The case study region is Haiti. The country, which has an extension of 27.750 $km^2$ is located in the Caribbean's Great Antilles and shares the island of Hispaniola with the Dominican Republic. The climate is predominantly tropical, with daily tempera-
tures ranging between 19°C and 28°C during winter and between 23°C and 33°C during summer. The island topography is varied; the central region is mainly mountainous, while the northern and western regions are near the coastline. Annual precip-itation in the central region averages 1.200 mm, while in the lowlands it is about 550 mm (GFDRR, 2011). Haiti is subject to the variability associated with El Niño and La Niña phenomena, with El Niño bringing drier and hotter conditions and La Niña colder and wetter climate. Haiti experiences a first rainy season from April to July and a second, and most important, from
August to the end of November. The dry season starts in December and goes on until the end of March (FEWSNET, 2019).

Haiti is divided administratively into 10 departments (Fig. 2), with people living mainly in the West, where the capital Port-au-Prince is located, and in the Artibonite. The total population in 2017 was about 11 million people (World Bank, 2017). Haiti is the poorest country in the Western Hemisphere, the economy is mainly agricultural. 67% of the country's area is devoted to agriculture, but only 4,35% of the agricultural area is irrigated (Trading Economics, 2013), posing a major threat to local
production.

Haiti produces over half of the world's vetiver oil (used in cosmetics), and mangos and cocoa are the most important export crops. Two-fifths of all Haitians depend on the agriculture sector, mainly small-scale subsistence farming. The country is prone to all types of natural hazards. Earthquakes, storms, hurricanes, landslides, and droughts have caused huge damages and losses in recent years. Haiti was ranked as the third most affected country by extreme weather events in terms of lives lost and
economic damages in the period from 1994 to 2013 (GFDRR, 2011). More than 96% of the population lives in areas at risk of two or more hazards. The most frequent disasters are floods and storms but, when considering the number of affected people, droughts are the disasters involving the highest number of persons (Fig. 3).

Droughts threat the livelihoods of Haitians in many different ways. The scarcity of crops production means a rise in food prices, that brings to widespread food insecurity since the major part of people can't afford the increase. Unavailability of
drinking water leads to cholera outbreaks among the population. Water is an issue also for breeders, who lose livestock on which they rely for milk production and meat consumption. In the period from 1980 to present more than 10 drought events have been reported by the government or the humanitarian organizations working in Haiti (Table 7). The worst drought was the one of 2014-2017, affecting more than 3 million inhabitants (about one-third of Haiti's population).

Effective drought management is crucial for Haiti, but at present, a reliable early warning system for drought is still lacking.
Weather stations on the ground are few and data records are often very short, therefore not useful for drought monitoring purposes on the entire country. Satellite images can be an effective and not expensive way to improve drought management and preparedness in the country.





## 3 Results and Discussion

### 3.1 Correlation analysis

Haiti has been divided into 987 grid cells, accounting for 90% of the country area. 1941 weeks were considered, starting from week 35 of 1981 and ending with week 52 of 2018. The release date of a new VHI image was considered as the starting date for a week. In the present study, four precipitation aggregation periods were considered (1 months, 2 months, 3 months and 6 months) and the corresponding values of SPI (SPI1, SPI2, SPI3 and SPI6) were computed in order to select the SPI aggregation timescale to be used to create the PPVI.

To evaluate the strength of the statistical relationship between the SPI at various timescales and the VHI a correlation analysis was then performed. Various studies have already evaluated the correlation among drought indices or between drought indices and exogenous variables; for example (Bonaccorso et al., 2015) investigated the correlation between SPI and NAO, while (Hongshuo et al., 2014) investigated the correlation between SPI (various aggregation periods) and the VHI. While in the majority of the papers the Pearson correlation coefficient was employed, in the present study the Spearman correlation 250 coefficient was preferred as a measure of the statistical relationship between the indices, as suggested in (Wedgbrow et al., 2002). The number of significant correlations at 5% and 1% was evaluated for four SPI aggregation timescales (Table 8).

The highest number of significant correlations was found in the cases of SPI2 and SPI3, which exhibit very similar performances at 1% significant level. This finding is in agreement with previous studies such as (Hongshuo et al., 2014), that found that VHI and SPI3 have the highest correlation for croplands, whereas VHI and 6-month SPI have the highest correlation for 255 forest in the Southwest of China; and (Ma'rufah et al., 2017) that found that significant correlation coefficient values on SPI3 and VHI are common in the southern part of Indonesia. Since SPI3 has been used in literature and the percentage of significant correlation at 1% level is relevant, it has been decided to aggregate SPI over a 3 months period and use SPI3 in the following discussion.

### 3.2 Normality of SPI and VHI distributions

Before computing PPVI as described in the previous sections, a test on the normality of the SPI3 and $VHI_{st}$ distributions was performed. The goodness of fit of the SPI3 and the $VHI_{st}$ distributions was verified through the histograms in Fig. 4 (panel (a) and (b) respectively), where the boxplots represent the relative frequencies of the SPI3 and $VHI_{st}$ values. Both the SPI3 and the $VHI_{st}$ data can therefore be considered normally distributed.

### 3.3 Selection of threshold values

PPVI was computed as described in Sect. 2.2 and its performance in identifying past drought events in Haiti when used in combination with the set of rules described in Sect. 2.2.4 was evaluated. To this end, the ROC curve classification methodology was applied. The set of rules implied that at first, cells in drought conditions were identified: drought started in a specific grid cell at week $W$ when PPVI was lower than the threshold $Z$ and ended when PPVI was up to the threshold $z$ in the same grid cell





at a week $w$ (with $w$ coming after $W$). Then a regional drought event was identified: the drought event started when more than

$N$ cells at a specific week $W_1$ were in drought conditions and ended at a week $W_2$ when few than $n$ grid cells were in drought conditions. The comparison was performed on a weekly basis, with observations derived from the reported events described in Table 7.

The ROC curves were computed according to the following methodology: at first a combination of the thresholds $Z$, $z$, $N$ and $n$ was selected. On the basis of the set of rules established in Sect. 2.2.4, the ability of the selected combination of thresholds

in reproducing the observations was assessed by computing $TP$, $TN$, $FP$ and $FN$ as defined in Table 6, together with POD and POFD. A couple (POFD, POD) represents a point in a ROC graph. Then one threshold among $Z$, $z$, $N$ and $n$ was selected. The selected threshold was variable during the analysis, while the other three were kept constant. The step of variation was identified according to the threshold maximum and minimum values. For each combination of the four thresholds (the varying one and the three fixed) $TP$, $TN$, $FP$, $FN$ and POD and POFD were computed. The resulting set of couples (POFD, POD)

represented the ROC curve for the considered set of thresholds.

The analysis was repeated by varying another threshold among $Z$, $z$, $N$ and $n$. As an example, Fig. 5 shows four ROC curves for the thresholds in Table 9. Thresholds $N$ and $n$ in Table 9 are expressed as the percentage of the country's area instead as the number of grid cells. For each of the curves the best performing set of ($Z$, $z$, $N$ and $n$) was selected by identifying the point farther from the 45°line, as done by (Zhu et al., 2016). The Area Under the Curve (AUC) was used as criteria to establish which

of the ROC curves should be preferred (as was done by (Dutra et al., 2014; Mason and Graham, 2002; Zhu et al., 2012)). An AUC near to 1 indicates good performance, while AUC of 0.5 indicates the model has no predictive skills. From Fig. 5 it is clear that the curve corresponding to the parameters defined as "Set 2" in Table 9 should be preferred, since the AUC is the closest to 1.

### 3.4    Indices comparison

The aim of this paragraph is not to validate in absolute terms the proposed methodology since the data record is too short to serve both for calibration and for validation. In the present section, instead, we provide a validation by comparing PPVI with widely recognized and used indices such as SPI and VHI.

The performance of PPVI was then compared to the one of SPI3 and VHI considered separately. Thresholds analogous to $Z$ and $z$ were defined for SPI3 and VHI. Thresholds $Z_S$ and $z_S$ mark respectively the beginning and the end of drought conditions

in a grid cell according to SPI3 and thresholds $Z_V$ and $z_V$ do the same in the case of VHI. Again the four thresholds $Z$, $z$, $N$, and $n$ were varied in order to identify the optimal values. As an example Fig. 6 shows a comparison among the ROC curves for the three indices. In each panel of Fig. 6, $n$ and $z$, $z_S$ and $z_V$ (for PPVI, SPI3 and VHI) remained constant, while $Z$, $Z_S$ and $Z_V$ were varying; $N$ was fixed in each panel but varied among the panels. $Z$ varied from -4 to -1.1 with a step equals to 0.1; $Z_S$ varied from -3 to 0 with a step equals to 0.1 and $Z_V$ varied from 10 to 40 with a step equal to 5.

It is clear from Fig. 6 that PPVI identified the reported drought events better than SPI3 and VHI. AUC was 0,828 for PPVI, 0,740 for SPI3 and 0,784 for VHI. The AUC value of PPVI was in line with similar results reported in literature (Mwangi et al., 2014). As can be seen from Fig. 6, the new index provided better results with respect to the ones obtained with SPI3 or VHI





considered separately. In all the four configurations shown in Fig. 6, the AUC for the curve constructed with PPVI was higher than the ones for SPI3 and VHI. The AUC values are in line with the ones considered good in the literature (see (Khadr, 2016))

for drought predictive skills. The optimal thresholds to configure the model were then determined by selecting the point farther from the 45°line, as done by (Zhu et al., 2016). The best configurations parameters are shown in Table 10 and comes from the PPVI curve shown in panel (c) of Fig. 6. The drought events were therefore identified using the optimal parameters (Table 10).

The ability of the model in identifying the country area hit by the drought was also assessed. A visual comparison among the area under drought identified by the three indices was performed, as was done by (Dutta and Kundu, 2015).

Here some significant weeks are shown. At first, week 45 of 1995 was considered. No drought events were reported in that period. Figure 7 shows that, while SPI3 identified all the southern part of the country as dry areas and VHI showed vegetation suffering in two departments (Centre and West), PPVI did not show signs of drought, except for a minor number of grid cells. Figure 8 shows that in 2015, when the whole country was reported to be in severe drought conditions, PPVI captured well the pattern, only a few grid cells were not in drought conditions. The SPI3 was also able to catch the situation, while for the VHI

only 58% of the county was in drought. During week 8 of 2012, only the Northern part of the country was in drought (Fig. 9); five departments were reported to be stressed (North, North West, North East, Artibonite, Centre, see Table 7).All the three indices showed the North West as the department most affected by drought when considering the percentage of the department area hit by the drought. PPVI then classified Artibonite, North, Centre and North East, while SPI3 as second and third most affected departments identified South and Grand Anse and VHI Centre and Nippes (Table 11).

Severity, duration and mean areal extent of the drought events identified by PPVI were computed. Severity was computed as the sum of all the values identified by the condition that a grid cell is in a drought condition when PPVI is lower than -1.8 and exits from drought when PPVI is up to -1.1. Duration is expressed in months and the mean areal extent is the average percentage of area in drought during a specific event. Results are presented in Table 12.

PPVI showed overall a better capacity in identifying drought events with respect to SPI3 and VHI considered separately.

However, some false alarms still remain. This can be linked to the uncertainty in information on past drought events for the analysed area. Short-term droughts are often not reported in text-based documents, and information on drought start and end date were retrieved from documents that mainly described the impacts related to drought. PPVI showed a good agreement with reported information in identifying the areas of the country hit by the drought.

## 4  Conclusions

The timely identification of drought events is of great importance in agricultural areas, especially when rainfed agriculture is practiced. At the same time, the evaluation of the damages caused by drought is a key point to select appropriate risk management strategies, such as weather index insurance programs, agricultural index insurance, disaster financing and early action planning. The new composite index proposed in this paper, the Probabilistic Precipitation Vegetation Index, PPVI, is a powerful tool since it can identify events of vegetation stress, and at the same time, select among those the ones actually due to

drought, thanks to the contemporary use of both VHI and SPI. As such it can be helpful in agricultural drought monitoring and



can be used to identify drought events affecting a region, their severity and their duration as was shown in the case of Haiti. In particular, PPVI can be precious in those areas where rainfed agriculture is of vital importance since people rely on it for food production for personal consumption.

Among the interesting aspects of PPVI, there is the fact that few data are required for its computation: only precipitation and the VHI. This aspect is crucial, since many composite indicators able to identify agricultural droughts already exist, but large amounts of data are required to compute them. For example, the United States Drought Monitor combines more than 40-50 inputs, while other indices specific for agricultural drought monitoring, such as the VegDRI and the VegOut, require the use of temperature and oceanic indices. The number of parameters required to compute PPVI is lower even with respect to OBDI, SWS, CDI or CDSI.

A second most important advantage is that, since the SPI was computed starting from satellite precipitation (CHIRP dataset) and that the VHI is a remote-sensing drought index, PPVI is also a remote-sensing product. The use of datasets with global coverage means that PPVI is easily transferable and scalable over the entire globe. In addition, PPVI can be a very useful tool in areas with scarce gauge coverage as the Caribbean Islands. Both precipitation and the VHI have a very high spatial and temporal resolution, thus allowing drought monitoring from satellite even in small areas. PPVI can be computed even in those regions with short data records, since the VHI has more than 30 years of records (data collection began in August 1981); and CHIRP precipitation are available from January 1981.

Both the SPI and the VHI are updated at weekly time-step since every week a new VHI image is released and the CHIRP precipitation dataset has a daily temporal resolution, therefore PPVI can be updated more frequently than other composite indices, such as CDI, which is updated every 10 days. In addition, due to the relatively short latency time (less than one week) of both the datasets employed to create PPVI, the index is available in near-real time, therefore allows the timely implementation of drought mitigation strategies. This last feature is of particular interest when PPVI is used to implement measure to reduce drought risk in agriculture, where a timely identification of drought is crucial to prevent damages to the sector.

Many advantages are also related to the adoption of the set of rules here proposed to identify drought events. First of all, these rules enable an objective and standardized identification of drought events from the mathematical point of view. Additionally, they can be adjusted according to the needs and the objectives of various possible end users of the model, such as farmers, governments or insurance companies.

The performances of PPVI in identifying drought events were tested in a specific case study (Haiti) and compared to the ones of SPI and VHI considered separately. PPVI performed better than the single indices considered separately in reproducing past drought events. PPVI identified drought areas in Haiti better than SPI and VHI even from the spatial point of view, thus it is more reliable than a single index. A comparison of PPVI performances with respect to the ones of other composite indices was not performed in the present study due to the unavailability of composite indices with the same characteristics of PPVI. In fact previous composite indices do not include both the meteorological and the agricultural aspect of drought or are not available globally, or cannot be computed with only remote sensing datasets.



*Author contributions.* This research is part of the PhD thesis from BM. BB and MM were both the thesis supervisors.

*Competing interests.* The authors declare that they have no conflict of interest.



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





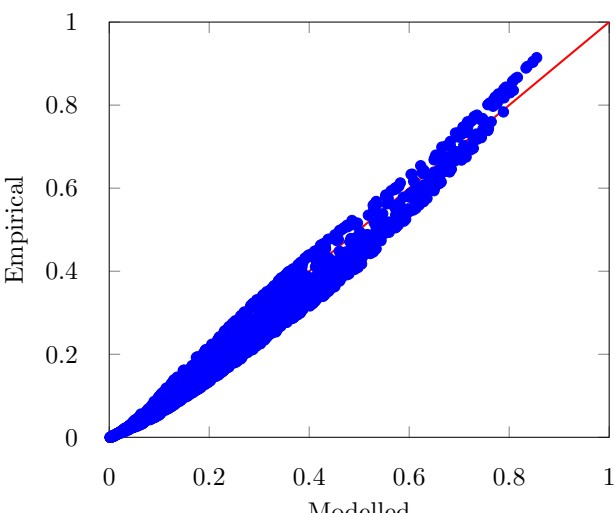

**Figure 1.** PPVI validation: empirical copula versus bivariate joint probability function. The red line corresponds to the 45°line. Joint probability values have been computed from Eq. 2, while empirical copula values according to Eq. 5.


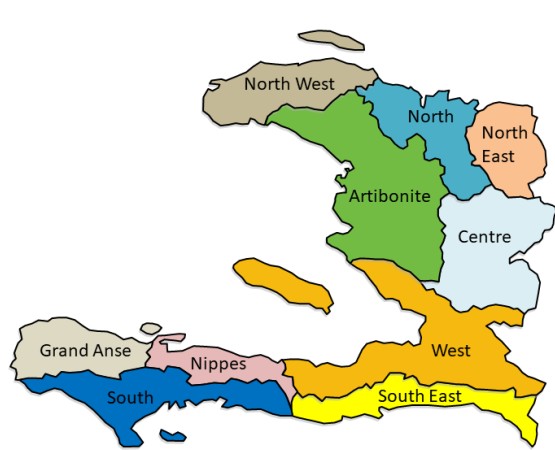

**Figure 2.** Map of Haiti departments.




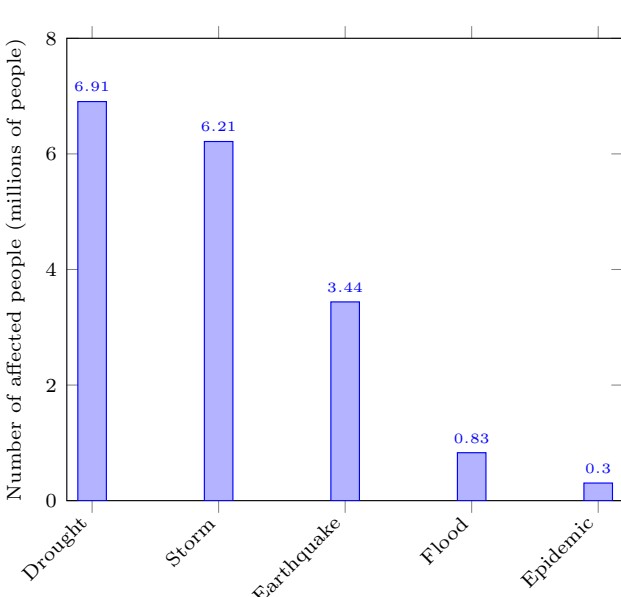

**Figure 3.** Number of people affected by natural disasters in Haiti (1900-2018). Source (CRED, 2017).

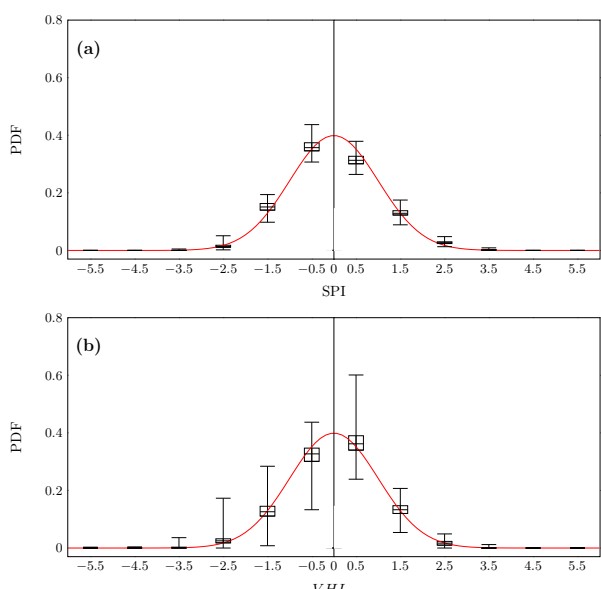

**Figure 4.** 4(a): distribution of $SPI$ values; 4(b):distribution of $VHI_{st}$ values. The red line represents the pdf of the standard normal distribution; boxplots represent the percentage of values lying in the range; 12 ranges were considered; starting from -6 and ending with 6 with a step equal to 1.

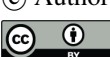



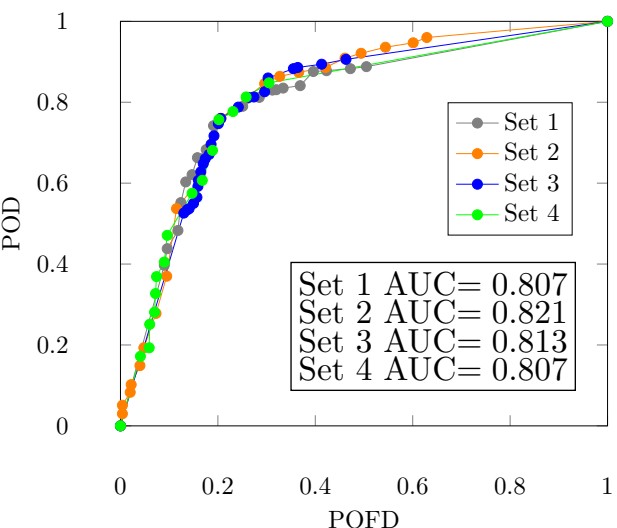

**Figure 5.** ROC curves for the set of thresholds reported in Table 9.





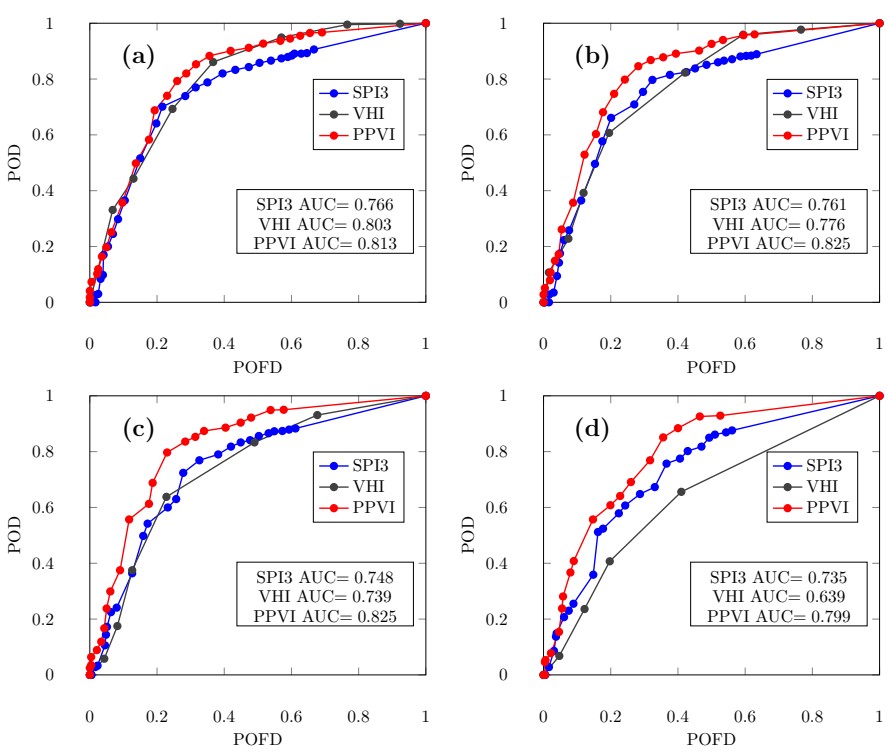

**Figure 6.** Comparison among the performances of SPI3, VHI and PPVI in identifying reported drought events; thresholds $Z$, $Z_S$ and $Z_V$ are varying, $z = -1.1$, $z_S = 0$ and $z_V = 40$; $n = 80$ and four cases for $N$ are shown: (a): $N = 10\%$, (b): $N = 20\%$; (c): $N = 30\%$ and (d): $N = 50\%$.



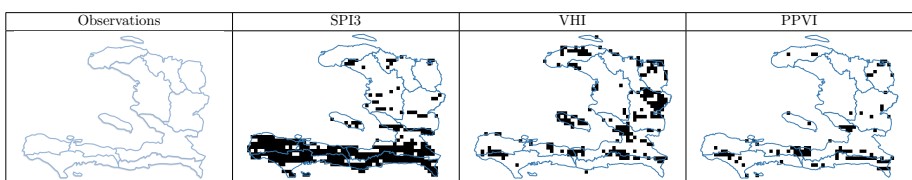

**Figure 7.** Comparison of the performance of SPI3, VHI, and PPVI in identifying the areas hit by drought. Week 45 of 1995. Departments highlighted in red are the ones in drought according to observations, black cells are the ones in drought condition according to the various indices.





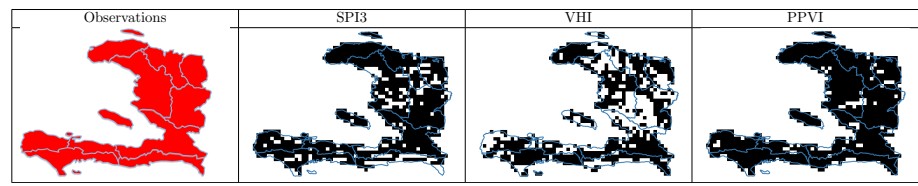

**Figure 8.** Same as Fig. 7 but for week 33 of 2015. Departments highlighted in red are the ones in drought according to observations, black cells are the ones in drought condition according to the various indices.




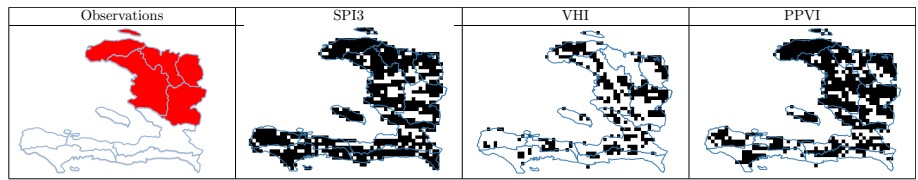

**Figure 9.** Same as Fig. 7 but for week 8 of 2012. Departments highlighted in red are the ones in drought according to observations, black cells are the ones in drought condition according to the various indices.




**Table 1.** An overview of indices used in agricultural drought monitoring; Met=Meteorological, Hydro=Hydrological, Ag=Agricultural.

| Index | Inputs | Drought type | Pros | Cons | Reference |
|---|---|---|---|---|---|
| ETDI | Modeled (SWAT) | Ag | Analysis of both actual and potential ET | Complex calculations | (Narasimhan and Srinivasan, 2005) |
| NDVI | Near-infrared | Ag | High resolution, global coverage, remote-sensing | Need for data processing | (Kogan, 1995a) |
| SMAI | Precipitation, Temperature, Available water content | Ag | Water balance approach | Data requirements | (Bergman et al., 1988) |
| SMDI | Modeled (SWAT) | Ag | Adaptable to different crop types | Based upon output from SWAT | (Narasimhan and Srinivasan, 2005) |
| SSI | Soil moisture | Ag | Uses soil moisture only, standardized | Based on one variable | (Hao and Aghakouchak, 2013) |
| SVI | VCI | Ag | Standardized, remote-sensing | Scarcerly employed | (Peters et al., 2002) |
| SWS | Available Water content, Rooting depth, Soil water deficit, Soil type | Ag | Well known calculations | Poor performance on non-homogeneous soils | (BC Ministry for Agriculture, 2015) |
| TCI | Brightness temperature | Ag | High resolution, global coverage, remote-sensing | Only brightness temperature is considered | (Kogan, 1995a) |
| VCI | NDVI | Ag | High resolution, global coverage, remote-sensing | Identifies all vegetation stresses, not only the ones due to drought | (Liu and Kogan, 1996) |
| CMI | Precipitation, Temperature | Ag | Weekly temporal resolution | Specifically developed for grain-producing regions in the USA | (Palmer, 1968) |





| Index | Inputs | Drought type | Pros | Cons | Reference |
|---|---|---|---|---|---|
| CSDI | Precipitation, Temperature, wind speed, solar radiation, dewpoint temperature, soil profile, plant phenology | Ag | Very specific for each crop, based on plant development | Many inputs with a daily temporal resolution | (Meyer et al., 1993) |
| CWSI | Actual and potential evapotranspiration | Ag | Useful for irrigation scheduling, remote-sensing | To be computed from MODIS data | (Idso et al., 1981) |
| NMDI | NDVI | Ag | Uses vegetation condition and soil water content | Poor performance in areas with sparse vegetation | (Wang and Qu, 2007) |
| RSM | Precipitation, Temperature, Evapotranspiration, Soil properties, Crop features, Crop management practice | Ag | Computes the water balance with various methods | Need for multiple inputs | (Thornthwaite and Mather, 1955) |
| DTx | Modeled (water balance) | Ag | Computes an integrated transpiration deficit over a period of time | Need for multiple inputs | (Matera et al., 2007) |
| ADI | Precipitation, Snow water content, Streamflow, Reservoir storage, Evapotranspiration, Soil water content | Met Hydro Ag | Water balance approach | Need for multiple inputs | (Keyantash and Dracup, 2004) |





**Table 2.** An overview of aggregate and composite drought indices useful for agricultural drought monitoring; Met=Meteorological, Hydro=Hydrological, Ag=Agricultural.

| Index | Inputs | Drought type | Pros | Cons | Reference |
|---|---|---|---|---|---|
| VegDRI | SPI, NDVI, PDSI | Ag | Use of surface and remote-sensing data | Short period of record, Available only for the contiguous USA | (Brown et al., 2008) |
| VHI | VCI, TCI | Ag | High temporal resolution, global coverage at a high spatial resolution, 30+ years of records | Identifies all types of vegetation stress, not only the drought-related ones | (Kogan, 1990) |
| MSDI | SPI, SSI | Met Ag | Global coverage, remote-sensing | Grid size may not represent all areas and climate regimes equally well; short period of records | (Hao and Aghakouchak, 2013) |
| CDI | SPI3, fAPAR, SMA | Met Ag | 10-day temporal resolution, high spatial resolution (5km) | Short period of record (2012), Available only in Europe, hard to replicate | (Sepulcre-Canto et al., 2012) |
| Morocco CDI | SPI, ET, LST, NDVI | Met Ag | High spatial resolution (5km) | Monthly temporal resolution; available for Morocco only, short period of record | (Bijaber et al., 2018) |



| Index | Inputs | Drought type | Pros | Cons | Reference |
|---|---|---|---|---|---|
| Hybrid DI | SPI, SWSI, PDSI | Met Hydro Ag | Function of damage, includes all types of droughts | Need for detailed information on economic damages | (Karamouz et al., 2009) |
| VegOut | SPI, NDVI, oceanic indices | Met Ag | Combination of climate information, vegetation condition, oceanic indices, and land cover | Need for a high number of parameters | (Tadesse and Wardlow, 2007) |
| OBDI | Precipitation, MPDI, Soil moisture | Met Ag | First attempt to combine and weight various inputs | Specifically designed for the USA | (Dieker et al., 2010) |
| USDM | PDSI, Soil moisture, Streamflow, Percent of normal precipitation, SPI, OBDI | Met Hydro Ag | Combines many inputs and expert knowledge, weekly temporal resolution | Available only for the USA | (Svoboda et al., 2002) |
| TVX | NDVI, LST | Ag | Combination of NDVI and temperature effects, remote-sensing | To be computed from NDVI and LST datasets | (Lambin and Ehrlich, 1995) |
| VTCI | NDVI, LST | Ag | Combination of NDVI and temperature effects, remote-sensing | To be computed from NDVI and LST datasets | (Wang et al., 2001) |
| AMDI-SA | MSPI, MSSI | Met Ag | Combination of SPI and SSI; standardized | Complex calculations | (Bateni et al., 2018) |





**Table 3.** Drought classification based on SPI according to (Mckee et al., 1993).

| Category | SPI | Probability (%) |
| --- | --- | --- |
| Extremely wet | 2.00 and above | 2.3 |
| Severely wet | 1.50 to 1.99 | 4.4 |
| Moderately wet | 1.00 to 1.49 | 9.2 |
| Near normal | -0.99 to 0.99 | 68.2 |
| Moderately dry | -1.49 to -1.00 | 9.2 |
| Severely dry | -1.50 to -1.99 | 4.4 |
| Extremely dry | -2 and below | 2.3 |



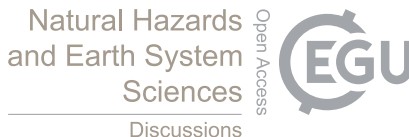

**Table 4.** Drought classification based on VHI according to (Dalezios et al., 2017).

| Category | VHI |
|---|---|
| Extremely dry | $\leq 10$ |
| Severely dry | $\leq 20$ |
| Moderately dry | $\leq 30$ |
| Mild dry | $\leq 40$ |
| Normal | $> 40$ |



**Table 5.** Drought classification according to PPVI.

| Category | PPVI | Probability (%) |
|---|---|---:|
| Extremely wet | 1.04 and above | 2.3 |
| Severely wet | 0.58 to 1.03 | 4.4 |
| Moderately wet | 0.13 to 0.57 | 9.2 |
| Near normal | -1.68 to 0.12 | 68.2 |
| Moderately dry | -2.14 to -1.69 | 9.2 |
| Severely dry | -2.15 to -2.59 | 4.4 |
| Extremely dry | -2.6 and below | 2.3 |





**Table 6.** Contingency table for the deterministic estimates of a series of binary events (Joliffe and Stephenson, 2012).

| Events estimated | Events Observed | | |
|---|---|---|---|
| | Yes | No | Total |
| Yes | $TP$ (True Positive or Hits) | $FP$ (False Positive or False Alarms) | $TP + FP$ |
| No | $FN$ (False Negative or Missing) | $TN$ (True Negative or Corret rejections) | $FN + TN$ |
| Total | $TP + FN$ | $FP + TN$ | $TP + FP + FN + TN = T$ |



**Table 7.** Reported drought events in Haiti from 1980 to present.

| Year | Department | Affected people | % population | Source |
|------|-----------|-----------------|--------------|--------|
| 1981 | South, Grand Anse, West | 103'000 | 2 | (Mora-Castro, 1986; CIAT, 2017) |
| 1982-1983 | South, South East, North West, North East | 333'000 | 5.75 | (Mora-Castro) |
| 1984-1985 | North West | 13'500 | 2 | (Mora-Castro, 1986; CIAT, 2017) |
| 1986 | All country | | | (Mora-Castro) |
| 1990-1992 | All country | 1'000'000 | 14 | (Mora-Castro) |
| 1997 | North West, North, North East | 50'000 | 0.64 | (CIAT, 2017) |
| 2000 | All counrty | | | (Mora-Castro) |
| 2003 | North West | 35'000 | 0.41 | (CIAT, 2017) |
| End 2009 | North West | | | (CNSA/MARNDR and FEWSNET, 2009) |
| 2011-2012 | North, North West, North East, Artibonite, Centre | | | (USAID et al., 2011; USAID and FEWSNET, 2012) |
| 2013 | All country | > 143'000 | 1.5 | (NOAA et al., 2013; FEWSNET, 2013) |
| 2014-2017 | All country | 3'600'000 | 33 | (OXFAM and Action contre la Faim, 2015; NOAA, 2017) |





**Table 8.** Number of significant correlations between VHI and various SPI aggregation timescales. Value is expressed as percentage evaluated with respect to the total number of grid cells (987).

|  | % significant correlations 5% | % significant correlations 1% |
|---|---|---|
| SPI1 | 94.53 | 90.78 |
| SPI2 | 97.26 | 95.44 |
| SPI3 | 96.66 | 95.34 |
| SPI6 | 89.77 | 85.61 |




**Table 9.** Example of set of thresholds used to draw ROC curves for model calibration. Thresholds $N$ and $n$ are expressed as the percentage of the country's area instead as the number of grid cells.

|  | $Z$ | $z$ | $N$ | $n$ | Step of variation |
|---|---|---|---|---|---|
| Set 1 | -2 | varying from -1.9 to 0 | 25% | 10% | 0.1 |
| Set 2 | varying from -3.5 to -1 | -1 | 25% | 10% | 0.1 |
| Set 3 | -2 | -1 | 25% | varying from 1% to 24% | 1% |
| Set 4 | -2 | -1 | varying from 11% to 25% | 10% | 1% |



**Table 10.** Best configuration parameters for the model when applied with PPVI.

| $Z$ | $z$ | $N$ | $n$ | TN | FP | FN | TP | POFD | POD |
|------|------|------|------|------|------|------|------|-------|-------|
| -1.8 | -1.1 | 300 | 80 | 957 | 379 | 99 | 506 | 0.284 | 0.836 |



**Table 11.** performance of PPVI, SPI3, and VHI in identifying departments hit by drought during week 8 of 2012 and comparison with observations.

| Department | Reported as drought | % of the area | | | Ranking of affected departments | | |
|---|---|---|---|---|---|---|---|
| | | PPVI | SPI3 | VHI | PPVI | SPI3 | VHI |
| North West | Yes | 93.1 | 91.7 | 47.2 | 1 | 1 | 1 |
| Artibonite | Yes | 75.1 | 72.8 | 34.1 | 2 | 7 | 5 |
| North | Yes | 74.6 | 82.1 | 10.4 | 3 | 4 | 9 |
| Centre | Yes | 67.2 | 54.3 | 45.7 | 4 | 10 | 2 |
| North East | Yes | 62.1 | 72.4 | 34.5 | 5 | 8 | 4 |
| West | No | 61.8 | 72.1 | 32.7 | 6 | 9 | 6 |
| Nippes | No | 51.2 | 75.6 | 36.6 | 7 | 5 | 3 |
| Grand Anse | No | 47.8 | 82.1 | 10.4 | 8 | 3 | 8 |
| South | No | 32.6 | 75.3 | 9 | 9 | 6 | 10 |
| South East | No | 30.8 | 84.6 | 20 | 10 | 2 | 7 |

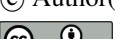



**Table 12.** Drought events in Haiti according to PPVI, duration, severity and mean areal extent.

| Event number | Start date | End date | Duration (months) | Mean intensity PPVI | Minimum PPVI | Mean areal extent (%) |
|---|---|---|---|---|---|---|
| 1 | 22/04/1982 | 14/07/1983 | 14 | -9.08 | -2.45 | 31.32 |
| 2 | 20/12/1984 | 07/11/1985 | 10 | -8.44 | -2.27 | 47.18 |
| 3 | 11/09/1986 | 09/04/1987 | 6 | -9.38 | -2.21 | 42.33 |
| 4 | 03/08/1989 | 18/10/1990 | 14 | -8.84 | -2.41 | 32.09 |
| 5 | 14/02/1991 | 04/02/1993 | 23 | -9.03 | -2.34 | 43.95 |
| 6 | 16/09/1993 | 27/01/1994 | 4 | -9.77 | -2.18 | 37.29 |
| 7 | 28/07/1994 | 10/11/1994 | 3 | -10.06 | -2.38 | 53.27 |
| 8 | 13/03/1997 | 15/01/1998 | 10 | -8.52 | -2.46 | 39.20 |
| 9 | 30/03/2000 | 28/09/2000 | 5 | -10.13 | -2.12 | 72.30 |
| 10 | 23/11/2000 | 26/04/2001 | 5 | -8.38 | -2.33 | 25.63 |
| 11 | 09/08/2001 | 13/12/2001 | 4 | -8.16 | -2.19 | 37.46 |
| 12 | 04/04/2002 | 20/06/2002 | 2 | -11.09 | -2.16 | 31.36 |
| 13 | 12/12/2002 | 30/10/2003 | 10 | -8.49 | -2.17 | 30.74 |
| 14 | 15/04/2004 | 27/05/2004 | 1 | -11.13 | -1.98 | 20.16 |
| 15 | 25/11/2004 | 19/05/2005 | 5 | -10.70 | -2.67 | 79.52 |
| 16 | 23/03/2006 | 13/07/2006 | 3 | -9.34 | -2.01 | 22.39 |
| 17 | 28/02/2008 | 31/07/2008 | 5 | -8.09 | -2.20 | 27.25 |
| 18 | 17/09/2009 | 18/02/2010 | 5 | -8.92 | -2.37 | 58.27 |
| 19 | 21/04/2011 | 16/06/2011 | 1 | -17.42 | -2.23 | 45.60 |
| 20 | 29/12/2011 | 05/04/2012 | 3 | -8.87 | -2.24 | 60.07 |
| 21 | 19/07/2012 | 01/11/2012 | 3 | -9.64 | -2.27 | 44.00 |
| 22 | 07/03/2013 | 05/05/2016 | 37 | -8.69 | -2.65 | 35.20 |
| 23 | 15/09/2016 | 20/04/2017 | 7 | -8.40 | -2.10 | 16.42 |
| 24 | 02/11/2017 | 14/12/2017 | 1 | -12.61 | -2.19 | 19.68 |
| 25 | 12/07/2018 | 31/12/2018 | 5 | -11.05 | -2.45 | 60.69 |