# Peer review of "A joint probabilistic index for objective drought identification: the case study of Haiti"

_Natural Hazards and Earth System Sciences, 2019_

## Referee Comment (RC1) · Anonymous Referee #1 · 3 Dec 2019

The manuscript 'nhess-2019-296' treats the interesting subject of objective drought identification by combining in a probabilistic framework two consolidated drought indices (SPI and VHI). Furthermore, the proposed index is a remote-sensing product since precipitations are retrieved from satellite and the VHI is a remote-sensing index. The manuscript can be published after improvement. The authors should take into consideration the following remarks:

Line 31: the two question marks should be deleted. Lines 248-251: The authors should explain why they used the Spearman correlation instead of the most common Pearson correlation coefficient. Of course, there is a reference concerning this subject (Wedgbrow et al. 2002) but this obliges the reader to find the reference in order to be informed.

Lines 300-301: 'It is clear from Fig.6 that PPVI identified the reported drought events better than SPI3 and VHI. AUC was 0,828 for PPVI, 0,740 for SPI3 and 0,784 for VHI.' We cannot observe the values 0,828 for PPVI, 0,740 for SPI3 and 0,784 for VHI referred in the figures.

Lines 326-328: 'Short-term droughts are often not reported in text-based documents, and information on drought start and end date were retrieved from documents that mainly described the impacts related to drought. PPVI showed a good agreement with reported information in identifying the areas of the country hit by the drought.' In Fig. 7, in the 'Observation' sub-figure, no department is highlighted in red. Does this mean that no drought was observed, or is this a mistake? In the former case, the authors should comment on this situation. In Figures 7 to 9 there is a comparison of indices and 'Observation' concerning the various departments of Haiti. Please define the criteria according to which a department is highlighted in red (drought conditions). Table 11. 'Reported as drought': Define the criteria of this classification.

Lines 365-366. A comparison of PPVI performances to the ones of other composite indices, would be a considerable improvement.

---

## Short Comment (SC1) · 3 Dec 2019

Drought is a naturally occurring event, which takes place in virtually all of the world's climatic regimes that results in significant economic, social, and environmental impacts in both developing and developed countries. Moreover, drought is widely recognized as a creeping natural hazard (Gillette, 1950) that occurs as temporary phenomena due to the natural climate variability. The increasing demand for food and water resources caused by a growing population and the potential to increase in severity, frequency and/or duration of droughts because of climate change have raised questions as to how humanity will withstand and confront future droughts. Consequently, drought is a serious problem throughout the world. Due to the multi-discipline character of the drought, a single, unique definition of a drought does not exist, but is subject to the

domain of interest of the observer (Niemeyer, 2008). The lack of general acceptance of a precise and objective definition of drought has been one of the principle obstacles to the investigation of drought (Yevjevich, 1967). Drought studies have been suffering from the lack of consistent methods for drought analysis. "Creeping" phenomena make detection of the onset and the end-point of a drought difficult to detect. Often these factors are determined long after the drought event has finished. For the identification, quantification and monitoring of drought phenomena, various methodologies have been proposed. The most popular of them seem to be single factors known as drought indices. Over the years, many drought indices were developed and used by meteorologists and climatologists around the world, which ranged from simple indices such as a percentage of normal precipitation and precipitation percentiles to more complicated indices such as the Palmer Drought Severity Index the Percent of Normal (PDSI), the Standardized Precipitation Index (SPI), and the Deciles and the Crop Moisture Index (CMI). However, no one of them has inherent priority over others, some of them have better performance in specific conditions. As Eslamian et al. (2017) pointed out, the drought monitoring using modern indices to predict the onset and termination of drought period, severity and other features is deemed necessary in order to develop the required measures to overcome the drought before its occurrence. The recent widespread and severe droughts that have resulted in serious economic, social, and environmental impacts in many countries have highlighted the need for improved drought monitoring. Hence, the manuscript investigates an important and timely environmental topic with global interest. Specifically, it suggests a new composite drought index, the Probabilistic Precipitation Vegetation Index (PPVI) based on the combination in a probabilistic framework two well-known drought indices, as the SPI (Standardized Precipitation Index) and the VHI (Vegetation Health Index) are. The PPVI presents some advantages such as: (a) Few data are required for its computation (precipitation and Vegetation Health Index); (b) It is a remote-sensing product; (c) It is easily transferable and scalable over the entire globe; (d) It can be a very useful tool in areas with scarce gauge coverage; (e) It is a powerful tool since it can identify events of vegetation

stress, and at the same time, select among those the ones actually due to drought; etc. This manuscript is well written, clear and well structured, follows a logical argumentation and restrictions of the results. My evaluation is positive. I suggest publishing the paper, but before the authors have to take into accounting my specific comments and remarks.

REFERENCES

Gillette, H. P.: A creeping drought under way, Water and Sewage Works, 104-105, l950.

Eslamian, S., Ostad-Ali-Askari, K., Singh, V.P. , Dalezios, N.R., Ghane, M., Yihdego, Y., Matouq, M. A.: Review of Drought Indices, Int J Constr Res Civ Eng (IJRCRE), 3(4), 48-66, http://dx.doi.org/10.20431/2454-8693.03040052017, 2017

Niemeyer, S.: New drought indices, in: Drought management: scientific and technological innovations, edited by:López-Francos, A., CIHEAM, Zaragoza, Spain, 267-274, (Options Méditerranéennes: Série A. Séminaires Méditerranéens; no. 80), . http://om.ciheam.org/om/pdf/a80/00800451.pdf, 2008

Yevjevich, V.: An objective approach to definition and investigations of continental hydrologic droughts, Hydrology papers, 23, Colorado State University, Fort Collins, USA, 1967

Please also note the supplement to this comment:
https://www.nat-hazards-earth-syst-sci-discuss.net/nhess-2019-296/nhess-2019-296-SC1-supplement.pdf
* * *
[Figure]

**Supplement:**

Drought is a naturally occurring event, which takes place in virtually all of the world's climatic regimes that results in significant economic, social, and environmental impacts in both developing and developed countries. Moreover, drought is widely recognized as a creeping natural hazard (Gillette, 1950) that occurs as temporary phenomena due to the natural climate variability. The increasing demand for food and water resources caused by a growing population and the potential to increase in severity, frequency and/or duration of droughts because of climate change have raised questions as to how humanity will withstand and confront future droughts. Consequently, drought is a serious problem throughout the world.

Due to the multi-discipline character of the drought, a single, unique definition of a drought does not exist, but is subject to the domain of interest of the observer (Niemeyer, 2008). The lack of general acceptance of a precise and objective definition of drought has been one of the principle obstacles to the investigation of drought (Yevjevich, 1967).

Drought studies have been suffering from the lack of consistent methods for drought analysis. "Creeping" phenomena make detection of the onset and the end-point of a drought difficult to detect. Often these factors are determined long after the drought event has finished.

For the identification, quantification and monitoring of drought phenomena, various methodologies have been proposed. The most popular of them seem to be single factors known as drought indices. Over the years, many drought indices were developed and used by meteorologists and climatologists around the world, which ranged from simple indices such as a percentage of normal precipitation and precipitation percentiles to more complicated indices such as the Palmer Drought Severity Index the Percent of Normal (PDSI), the Standardized Precipitation Index (SPI), and the Deciles and the Crop Moisture Index (CMI). However, no one of them has inherent priority over others, some of them have better performance in specific conditions.

As Eslamian et al. (2017) pointed out, the drought monitoring using modern indices to predict the onset and termination of drought period, severity and other features is deemed necessary in order to develop the required measures to overcome the drought before its occurrence. The recent widespread and severe droughts that have resulted in serious economic, social, and environmental impacts in many countries have highlighted the need for improved drought monitoring.

Hence, the manuscript investigates an important and timely environmental topic with global interest. Specifically, it suggests a new composite drought index, the Probabilistic Precipitation Vegetation Index (PPVI) based on the combination in a probabilistic framework two well-known drought indices, as the SPI (Standardized Precipitation Index) and the VHI (Vegetation Health Index) are.

The PPVI presents some advantages such as: (a) Few data are required for its computation (precipitation and Vegetation Health Index); (b) It is a remote-sensing product; (c) It is easily transferable and scalable over the entire globe; (d) It can be a very useful tool in areas with scarce gauge coverage; (e) It is a powerful tool since it can identify events of vegetation stress, and at the same time, select among those the ones actually due to drought; etc.

This manuscript is well written, clear and well structured, follows a logical argumentation and restrictions of the results. My evaluation is positive. I suggest publishing the paper, but before the authors have to take into accounting my specific comments and remarks.

**REFERENCES**

Gillette, H. P.: A creeping drought under way, Water and Sewage Works, 104-105, l950.

Eslamian, S., Ostad-Ali-Askari, K., Singh, V.P., Dalezios, N.R., Ghane, M., Yihdego, Y., Matouq, M. A.: Review of Drought Indices, Int J Constr Res Civ Eng (IJRCRE), 3(4), 48-66, http://dx.doi.org/10.20431/2454-8693.03040052017, 2017

Niemeyer, S.: New drought indices, in: Drought management: scientific and technological innovations, edited by:López-Francos, A., CIHEAM, Zaragoza, Spain, 267-274, (Options Méditerranéennes: Série A. Séminaires Méditerranéens; no. 80), . http://om.ciheam.org/om/pdf/a80/00800451.pdf, 2008

Yevjevich, V.: An objective approach to definition and investigations of continental hydrologic droughts, Hydrology papers, 23, Colorado State University, Fort Collins, USA, 1967

**Specific comments and remarks**

**Lines 17 & 18**

"World Meteorological Organizations (MWO)" should be "World Meteorological Organization (MWO)"

**Line 32**

"…. subsurface water supply??" **COMMENT**: Please, clarify what does it mean the term "subsurface water supply"?

**Line 40**

"….such as in (Gu et al., 2008). Many…." should be "….such as in Gu et al. (2008). Many…."

**Line 52**

Please define the abbreviation "VegDRI"

**Line 55**

"…Drought Indicators (OBDI)…" should be "…Drought Indicator (OBDI)…"

**Line 59**

"… as in (Serinaldi et al., 2009) and (Bonaccorso et al., 2012), where…"

should be

"… as in Serinaldi et al. (2009) and Bonaccorso et al. (2012), where…"

**Line 60**

"….or in (Shiau, 2006), where…" should be "….or in Sh iau (2006), where…"

**Line 61**

"….in Taiwan. (Shiau et al., 2007) investigates…" should be "….in Taiwan. Shiau et al. (2007) investigates…"

**Lines 62 & 63**

"….is used in (Songbai and Singh, 2010) to model…" should be "….is used in Songbai and Singh (2010) to model…"

**Line 68**

Please define the abbreviation "AMDI-SA"

**Line 78**

"…Vegetation Temeprature Condition…" should be "…Vegetation Temperature Condition…"

**Line 110**

"….described in (Funk et al., 2015). In the present…" should be "….described in Funk et al. 2015). In the present…"

**Line 130**

"….as proposed in (Mckee et al., 1993), is reported…." should be "….as proposed in Mckee et al. (1993), is reported…."

**Line 140**

"…as proposed in (Dalezios et al., 2017), is presented…" should be "…as proposed in Dalezios et al. (2017), is presented…"

**Lines 152 & 153**

"….proposed in (USDA Risk Management Agency et al., 2006)," should be "….proposed in USDA Risk Management Agency et al. (2006),"

**Lines 156 & 157**

"…as it is defined by (Kotz et al., 2000). The normality…" should be "…as it is defined by Kotz et al. (2000). The normality…"

**Line 164**

"…according to Eq. 3 and Eq.4 respectively, where…" should be "…according to Eqs. 3 and 4 respectively, where…"

**Line 168**

"…as done in (Kao and Govindaraju, 2010). The bivariate…" should be "…as done in Kao and Govindaraju (2010). The bivariate…"

**Line 169**

"….according to (Nelsen, 2006) using the…" should be "….according to Nelsen (2006) using the…"

**Line 188**

"….described in (Joliffe and Stephenson, 2012). The ROC…." should be "….described in Joliffe and Stephenson (2012). The ROC…."

**Line 198**

"…according to (Joliffe and Stephenson, 2012) with the…" should be "…according to Joliffe and Stephenson (2012) with the…"

**Line 247**

Please define the abbreviation "NAO"

**Line 248**

"…while (Hongshuo et al., 2014) investigated the…" should be "…while Hongshuo et al.(2014) investigated the…"

**Lines 250 & 251**

"….as suggested in (Wedgbrow et al., 2002). The number…." should be "….as suggested in Wedgbrow et al. (2002). The number…."

**Line 253**

"…studies such as (Hongshuo et al., 2014), that found…" should be "…studies such as Hongshuo et al. (2014), that found…"

**Line 256**

"…and (Ma'rufah et al., 2017) that found that significant correlation.." should be "…and Ma'rufah et al. (2017) found that significant correlation.."

**Line 284**

"…as done by (Zhu et al., 2016). The Area…" should be "…as done by Zhu et al. (2016). The Area…"

**Line 285**

"…should be preferred (as was done by (Dutra et al., 2014; Mason and Graham, 2002; Zhu et al., 2012)). An…" should be "…preferred as was done by Dutra et al. (2014); Mason and Graham (2002); Zhu et al. (2012). An…"

**Lines 304 & 305**

"…good in the literature (see (Khadr, 2016)) for drought… " should be "…good in the literature (Khadr, 2016) for drought… "

**Line 306**

"…as done by (Zhu et al., 2016). The best…" should be "…as done by Zhu et al. (2016). The best…"

**Line 309**

"…as was done by (Dutta and Kundu, 2015)…." should be "…as was done by Dutta and Kundu (2015)…."

**Lines 405-407**

Dutta, D. and Kundu, A.: Assessment of agricultural drought in Rajasthan ( India ) using remote sensing derived Vegetation Condition Index ( VCI ) and Standardized Precipitation Index (SPI), The Egyptian Journal of Remote Sensing and Space Sciences, 18, 53–63, https://doi.org/10.1016/j.ejrs.2015.03.006, http://dx.doi.org/10.1016/j.ejrs.2015.03.006, 2015.

Should be

Dutta, D., Kundu, A, Patel, N.R., Saha, S.K., and iddiqui, A.R.: Assessment of agricultural drought in Rajasthan (India) using remote sensing derived Vegetation Condition Index (VCI) and Standardized Precipitation Index (SPI), The Egyptian Journal of Remote Sensing and Space Sciences, 18, 53-63, http://dx.doi.org/10.1016/j.ejrs.2015.03.006, 2015

**Lines 421 & 422**

Hao, Z.: Review of dependence modeling in hydrology and water resources, Progress in Physical Geography, 40, 549–578, https://doi.org/10.1177/0309133316632460, 2016.

should be

Hao, Z. and Singh, V.P.: Review of dependence modeling in hydrology and water resources, Progress in Physical Geography, 40, 549–578, https://doi.org/10.1177/0309133316632460, 2016.

**Line 489**

Palmer, W.C.: Keeping Track of Crop Moisture Conditions, Nationwide: The New Crop Moisture Index, Weatherwise, 21, 1968

should be

Palmer, W.C.: Keeping Track of Crop Moisture Conditions, Nationwide: The New Crop Moisture Index, Weatherwise, 21, 156-161, https://doi.org/10.1080/00431672.1968.9932814, 1968

**Line 496**

Shiau, J.-T.: Fitting Drought Duration and Severity with Two-Dimensional Copulas, Water Resources Management, 20, 795–815, 2006.

should be

Shiau, J.-T.: Fitting Drought Duration and Severity with Two-Dimensional Copulas, Water Resources Management, 20, 795–815, https://doi.org/10.1007/s11269-005-9008-9, 2006.

---

## Referee Comment (RC2) · Anonymous Referee #2 · 26 Dec 2019

This paper proposes a new composite drought index that accounts for both meteorological and agricultural drought conditions, by combining in a probabilistic framework two consolidated drought indices: the Standardized Precipitation Index (SPI) and the Vegetation Health Index (VHI). The new index, called Probabilistic Precipitation Vegetation Index (PPVI), is scalable, transferable all over the globe and can be updated in near-real time. is focused on the extremeness of recent drought events in Switzerland by looking at different types of drought, including meteorological, hydrological, agricultural, and groundwater drought. The paper is a new research study and is generally well-written as it explains the methodology, the mathematical framework and the assumptions used. However, the application research part needs minor improvements and corrections to verify the novelties of the method employed in the study area. Based

on this general comment the following points should be addressed and clarified.

1. Use of CHIRP dataset instead of the CHIRPS dataset. Please justify why the CHIRP dataset is used for the study area. Based on the study of Funk et al., 2015 it is proved that constraining the CHIRP by the CHPclim reduces systematic estimation errors and the CHIRPS dataset produces low MAE and bias statistics than the CHIRP dataset. It would be interesting to see a comparison of the observed precipitation pattern with the used rainfall datasets. How close is the used dataset with the observed rainfall in Haiti? Please provide scientific evidence in the revised manuscript which demonstrates the superiority of the used dataset when compared with the CHIRPS dataset for spatial and temporal (monthly) rainfall modelling at the study area.

2. Vegetation Health Index: It should be mentioned that all remote sensing indices could be expressed as deviations from the mean using the standardization procedure (i.e Mckee et al., 1993) as used by Peters et al. [2002]. Hence, the adopted classification of VHI using Eq. 1 is a transformation procedure of the typical VHI (from 0 to 1) to a normal distribution using the standardization procedure as proposed by Peters et al., [2002]. I recommend to the authors to clarify this issue on the revised manuscript and to mention that equal weighting is used for VCI and TCI. Furthermore, please discuss why VHI is used using the approach of Kogan and why VCI and TCI and are not first standardized and then combined with equal [??? see also Bento et al., 2018a,b] weighting in a probabilistic form to give the VHI (similar approach to PPVI or the approach of multivariate distributions using parametric [Hao and AghaKouchak, 2013] or a non-parametric approaches [Hao and AghaKouchak, 2014]).

3. Comparison with identified drought events. Is it possible to include a section with a comparison of PPVI with historical identified drought events? This comparison could exemplify the proposed index and strengthen the scientific quality of the manuscript.

For the motivations listed above, the paper in its present form needs revisions in order to evaluate the innovative character of the proposed method. The paper is of general

interest for international audience and merits publication in NHESS journal when the revisions and comments are addressed. Addressing these comments will improve the quality of the paper and help the general reader of the paper.

Bento, V.A.; Gouveia, C.M.; DaCamara, C.C.; Trigo, I.F. A climatological assessment of drought impact on vegetation health index. Agric. For. Meteorol. 2018a, 259, 286–295.

Bento, V.A.; Trigo, I.F.; Gouveia, C.M.; DaCamara, C.C. Contribution of Land Surface Temperature (TCI) to Vegetation Health Index: A Comparative Study Using Clear Sky and All-Weather Climate Data Records. Remote Sens. 2018b, 10, 1324.

Funk, C., Peterson, P., Landsfeld, M., Pedreros, D., Verdin, J., Shukla, S., Husak, G., Rowland, J., Harrison, L., Hoell, A., andMichaelsen, J.: The climate hazards infrared precipitation with stations - A new environmental record for monitoring extremes, Scientific Data, 2, 1–21, https://doi.org/10.1038/sdata.2015.66, 2015.

Hao Z., AghaKouchak A. Multivariate Standardized Drought Index: A Parametric Multi-Index Model, Advances in Water Resources, 57, 12-18, 2013, doi: 10.1016/j.advwatres.2013.03.009.

Hao Z., AghaKouchak A. A Nonparametric Multivariate Multi-Index Drought Monitoring Framework, Journal of Hydrometeorology, 15, 89-101, 2014, doi:10.1175/JHM-D-12-0160.1.

Mckee, T. B., Doesken, N. J., and Kleist, J.: The relationship of drought frequency and duration to time scales, in: AMS 8th Conference on Applied Climatology, January, pp. 179–184, 1993.

Peters, A.J., Walter-Shea, E.A., Ji, L., Vina, A., Hayes, M. and Svoboda, M.D., 2002. Drought monitoring with NDVI-based standardized vegetation index. Photogrammetric engineering and remote sensing, 68(1), pp.71-75.

2019-296, 2019.

---

## Author Comment (AC1) · 16 Jan 2020

We would like to thank Referee #2 for the insightful comments on the paper. Detailed responses to each comment together with the changes that will be done in the revised version of the manuscript are in the attached supplement. RC indicates the referee comments, AC the authors comments. Where useful, we reported both the original manuscript's text and the authors changes to the manuscript to help referees to visualize the changes that will be done in the revised version of the manuscript.

Please also note the supplement to this comment:
https://www.nat-hazards-earth-syst-sci-discuss.net/nhess-2019-296/nhess-2019-296-AC1-supplement.pdf

---

## Author Comment (AC2) · 16 Jan 2020

The authors would like to thank Dr. Stavros Yannopoulos for his positive feedback on the manuscript. All his remarks were useful to improve the paper readability and have been addressed in the revised version of the manuscript.

---

## Author Response (AR1)

**Authors response to Referee #1**

We thank Referee #1 for the feedback provided. All the comments and the suggestions were useful to improve the quality and the readability of the paper. Detailed responses to each comment are given below together with the changes that will be done in the revised version of the manuscript. RC indicates the referee comments, AC the authors comments. To help referees to visualize the changes that will be done in the revised version of the manuscript we reported both the original manuscript's text and the authors changes to the manuscript.

**RC1:** Line 31: the two question marks should be deleted

**AC:** it's a mistake due to a missing reference.

**Original manuscript**: Meteorological drought is related to precipitation shortages; hydrological drought refers to periods of precipitation shortfall on surface or subsurface water supply ??, while agricultural drought is conventionally linked to soil moisture deficit.

**Author's changes to the manuscript (Lines 31-32)**: Meteorological drought is related to precipitation shortages; hydrological drought refers to periods of precipitation shortfall on surface or subsurface water supply (Sheffield & Wood, 2011) while agricultural drought is conventionally linked to soil moisture deficit.

**RC1**: Line 248-251: The authors should explain why they used the Spearman correlation instead of the most common Pearson correlation coefficient. Of course, there is a reference concerning this subject (Wedgbrow et al., 2002) but this obliges the reader to find the reference in order to be informed.

**AC**: We have reconsidered the use of the Pearson correlation coefficient instead of Spearman. In fact, the use of the Pearson correlation coefficient could be preferable when dealing with normal variables. Therefore, the Spearman correlation coefficient has been substituted with the Pearson correlation coefficient. The manuscript will be changed accordingly.

**Original manuscript:** While in the majority of the papers the Pearson correlation coefficient was employed, in the present study the Spearman correlation coefficient was preferred as a measure of the statistical relationship between the indices, as suggested in (Wedgbrow et al., 2002). The number of significant correlations at 5% and 1% was evaluated for four SPI aggregation timescales (Table 8). The highest number of significant correlations was found in the cases of SPI2 and SPI3, which exhibit very similar performances at 1% significant level. This finding is in agreement with previous studies such as (Hongshuo et al., 2014) that found that VHI and SPI3 have the highest correlation for croplands, whereas VHI and 6-month SPI have the highest correlation for forest in the Southwest of China; and (Ma'rufah, 2017) that found that significant correlation coefficient values on SPI3 and VHI are common in the southern part of Indonesia. Since SPI3 has been used in literature and the percentage of significant correlation at 1% level is relevant, it has been decided to aggregate SPI over a 3 months period and use SPI3 in the following discussion.

*Table 8: Number of significant correlations between VHI and various SPI aggregation timescales. Value is expressed as percentage evaluated with respect to the total number of grid cells (987).*

|  | % significant correlations 5% | % significant correlations 1% |
|---|---|---|
| SPI1 | 94.53 | 90.78 |
| SPI2 | 97.26 | 95.44 |
| SPI3 | 96.66 | 95.34 |
| SPI6 | 89.77 | 85.61 |

**Author's changes to the manuscript (Lines 248-259):** The Pearson correlation coefficient was employed in the present study as a measure of the statistical relationship between the indices. The number of significant correlations at 5% and 1% was evaluated for four SPI aggregation timescales (Table 8). The highest number of significant correlations was found in the cases of SPI2 and SPI3, which exhibit very similar performances at 1% significant level. This finding is in agreement with previous studies such as (Hongshuo et al., 2014) that found that VHI and SPI3 have the highest correlation for croplands, whereas VHI and 6-month SPI have the highest correlation for forest in the Southwest of China; and (Ma'rufah, 2017) that found that significant correlation coefficient values on SPI3 and VHI are common in the southern part of Indonesia. Since SPI3 has been used in literature and the percentage of significant correlation at 1% level is relevant, it has been decided to aggregate SPI over a 3 months period and use SPI3 in the following discussion.

*Table 8: Number of significant correlations (Pearson correlation coefficient) between VHI and various SPI aggregation timescales. Value is expressed as percentage evaluated with respect to the total number of grid cells (987).*

|      | % significant correlations 5% | % significant correlations 1% |
|------|-------------------------------|-------------------------------|
| SPI1 | 93.52                         | 91.29                         |
| SPI2 | 96.76                         | 95.34                         |
| SPI3 | 96.15                         | 94.83                         |
| SPI6 | 90.07                         | 85.82                         |

**RC1:** Line 300-301: "It's clear from Fig.6 that PPVI identified the reported drought events better than SPI3 and VHI. AUC was 0.828 for PPVI, 0.740 for SPI3 and 0.784 for VHI." We cannot observe the values 0.828 for PPVI, 0.740 for SPI3 and 0.784 for VHI referred in the figures.

**AC:** The sentence can be rephrased, and the Figure 6 can be adjusted as follows.

**Original manuscript:** It's clear from Fig.6 that PPVI identified the reported drought events better than SPI3 and VHI. AUC was 0.828 for PPVI, 0.740 for SPI3 and 0.784 for VHI.

**Author's changes to the manuscript (Lines 300-304):** It's clear from Fig. 6 that the red curve, representing PPVI, is the furthest from the diagonal line in all the panels of the figure. The Area Under the Curve (AUC) was used as criteria to establish which index gave the best performances. AUC values are shown in Fig. 6 for each index and various configurations of the model.

[Figure]

*Figure 6: Comparison among the performances of SPI3, VHI and PPVI in identifying reported drought events; thresholds Z, $Z_S$ and $Z_V$ are varying, $z = -1.1$, $z_s = 0$ and $z_v = 40$; n = 80 and four cases for N are shown: (a): N = 10%; (b): N = 20%; (c):N = 30% and (d): N=50%.*

**RC1:** Lines 326-328: 'Short-term droughts are often not reported in text-based documents, and information on drought start and end date were retrieved from documents that mainly described the impacts related to drought. PPVI showed a good agreement with reported information in identifying the areas of the country hit by the drought.' In Fig. 7, in the 'Observation' sub-figure, no department is highlighted in red. Does this mean that no drought was observed, or is this a mistake? In the former case, the authors should comment on this situation. In Figures 7 to 9 there is a comparison of indices and 'Observation' concerning the various departments of Haiti. Please define the criteria according to which a department is highlighted in red (drought conditions). Table 11. 'Reported as drought': Define the criteria of this classification.

**AC:** In Fig.7 no department was highlighted in red since no drought was observed during that week according to text-based documents regarding droughts in Haiti. Departments are highlighted in red if, according to the documents cited in Table 7, drought was observed during that week in the department. The same criteria were adopted in Table 11 to establish if, according to observations, a department was in drought.

Figure 7, 8 and 9 will be modified to include a legend to clearly distinguish between departments in drought and departments not in drought. A description of the criteria used to define drought according to observation

will be given. The text of the manuscript will be modified to clarify the criteria adopted to identify drought in the various departments.

**Author's changes to the manuscript: (Lines 310 – 319):** At first, week 45 of 1995 was considered. No drought events were reported in that period according to the information available in the analysed documents (see Table7). Figure 7 shows that, while SPI3 identified all the southern part of the country as dry areas and VHI showed vegetation suffering in two departments (Centre and West), PPVI did not show signs of drought, except for a minor number of grid cells. Figure 8 shows that in 2015, when the whole country was reported to be in severe drought conditions (see Table 7 and (NOAA, 2017; OXFAM & Action conte la Faim, 2015)), PPVI captured well the pattern, only a few grid cells were not in drought conditions. The SPI3 was also able to catch the situation, while for the VHI only 58% of the county was in drought. During week 8 of 2012, only the Northern part of the country was in drought (Fig. 9), as highlighted by (USAID & FEWSNET, 2012) (see Table 7). Five departments were reported to be stressed (North, North West, North East, Artibonite, Centre, see Table 7). All the three indices showed the North West as the department most affected by drought when considering the percentage of the department area hit by the drought. PPVI then classified Artibonite, North, Centre and North East, while SPI3 as second and third most affected departments identified South and Grand Anse and VHI Centre and Nippes (Table 11).

[Figure]

Figure 7: Comparison of the performance of SPI3, VHI, and PPVI in identifying the areas hit by drought. Week 45 of 1995. Departments highlighted in red are the ones in drought according to observations (Table 7), red cells are the ones in drought condition according to the various indices.

[Figure]

Figure 8: Same as Fig. 7 but for week 33 of 2015. Departments highlighted in red are the ones in drought according to observations, red cells are the ones in drought condition according to the various indices.

[Figure]

Figure 9: Same as Fig. 7 but for week 8 of 2012. Departments highlighted in red are the ones in drought according to observations, red cells are the ones in drought condition according to the various indices.

*Table 11. Performance of PPVI, SPI3, and VHI in identifying departments hit by drought during week 8 of 2012 and comparison with observations. Observations are retrieved from the text-based documents reported in Table 7.*

| | | % of the area | | | Ranking of affected departments | | |
| --- | --- | --- | --- | --- | --- | --- | --- |
| Department | Reported as in drought | PPVI | SPI3 | VHI | PPVI | SPI3 | VHI |
| North West | Yes | 93.1 | 91.7 | 47.2 | 1 | 1 | 1 |
| Artibonite | Yes | 75.1 | 72.8 | 34.1 | 2 | 7 | 5 |
| North | Yes | 74.6 | 82.1 | 10.4 | 3 | 4 | 9 |
| Centre | Yes | 67.2 | 54.3 | 45.7 | 4 | 10 | 2 |
| North East | Yes | 62.1 | 72.4 | 34.5 | 5 | 8 | 4 |
| West | No | 61.8 | 72.1 | 32.7 | 6 | 9 | 6 |
| Nippes | No | 51.2 | 75.6 | 36.6 | 7 | 5 | 3 |
| Grand Anse | No | 47.8 | 82.1 | 10.4 | 8 | 3 | 8 |
| South | No | 32.6 | 75.3 | 9 | 9 | 6 | 10 |
| South East | No | 30.8 | 84.6 | 20 | 10 | 2 | 7 |

**RC1:** A comparison of PPVI performances to the ones of other composite indices, would be a considerable improvement.

**AC:** As already discussed in the manuscript (lines 365-368), a comparison with other composite indices is hard, due to the unavailability of composite indices with the same characteristics of PPVI. In fact, previous composite indices do not include both the meteorological and the agricultural aspect of drought or are not available globally or cannot be computed with only remote sensing datasets. In addition, VHI is already a composite drought index since it is derived from the linear combination of TCI and VCI. Therefore, in the manuscript, a comparison of PPVI performance with respect to the ones of a composite drought index was already performed.

**Authors response to Referee #2**

We would like to thank Referee #2 for the insightful comments on the paper. Detailed responses to each comment are given below together with the changes that will be done in the revised version of the manuscript. RC indicates the referee comments, AC the authors comments. To help referees to visualize the changes that will be done in the revised version of the manuscript we reported both the original manuscript's text and the authors changes to the manuscript.

**RC2**: Use of CHIRP dataset instead of the CHIRPS dataset. Please justify why the CHIRP dataset is used for the study area. Based on the study of Funk el al., 2015 it is proved that constraining the CHIRP by the CHPclim reduces systematic errors and the CHIRPS dataset produces low MAE and bias statistic than the CHIRP dataset.

**AC**: The authors are aware of the biases in the CHIRP dataset, but they aim at proposing an index for drought monitoring in near-real time; therefore, they selected the product with the shortest latency time. The main reason for using CHIRP instead of CHIRPS is the reduced latency time of the first dataset. In fact, as reported by (Funk et al., 2015), CHIRPS latency time is about three weeks; a preliminary version of CHIRPS with a 2-day latency time is available for GTS and Mexico only. In the case of CHIRP, latency time is about 2 days, and the product is available all over the world. CHIRP latency time can be checked at (Climate Hazard Group, 2015) by looking at the availability of the images.

**Original manuscript:** The use of CHIRP instead of CHIRPS (the Climate Hazard Group Infrared Precipitation with Stations) is related to the data latency time, which is shorter in the case of CHIRP since it doesn't include data from weather stations.

**Author's changes to the manuscript (lines 107-111)**: The use of CHIRP instead of CHIRPS (the Climate Hazard Group Infrared Precipitation with Stations) is related to the data latency time. Since the aim of the work is the development of an index for near-real time drought monitoring, the product with the shortest latency time was selected. CHIRPS data have a latency time of about three weeks (Funk et al., 2015), while CHIRP's latency is about 2 days, as can be checked on the dataset website (Climate Hazard Group, 2015)

**RC2**: It would be interesting to see a comparison of the observed precipitation pattern with the used rainfall dataset. How close is the used dataset with the observed rainfall in Haiti? Please, provide scientific evidence in the revised manuscript which demonstrates the superiority of the used dataset when compared with the CHIRPS dataset for spatial and temporal (monthly) rainfall modelling at the study area.

**AC**: Unfortunately, as highlighted by (Mari et al., 2015) "Systematic collection of rainfall through rain gauges has been relatively rare in post-earthquake Haiti, with on-the-ground rainfall measurements available only for Ouest (by USGS) and Sud (by Haiti Regeneration Initiative) departments". A map of existing rain gauges in Haiti reported in Eisenberg et al. (2013) shows the presence of only 5 gauges all over the country recording for a short period of time. Thus, a comparison between observed precipitation and rainfall retrieved from satellite images is not very feasible. The authors tried to overcome the issue providing a comparison with observed drought events, retrieved from text-based documents and international disasters databases, in Section 3.4 of the manuscript.

**RC2**: Vegetation Health Index: It should be mentioned that all remote sensing indices could be expressed as deviations from the mean using the standardization procedure (i.e. Mckee at al., 1993) as used by Peters et al. [2002]. Hence the adopted classification of VHI using Eq. 1 is a transformation procedure of the typical VHI (from 0 to 1) to a normal distribution using the standardization procedure as proposed by Peters et al.,[2002]. I recommend to the authors to clarify this issue on the revised manuscript and to mention that equal weighting is used for VCI and TCI.

**AC**: All these comments will be addressed in the revised manuscript as follows.

**Original manuscript:** The VHI is a remote-sensing index developed to include the effects of temperature on vegetation; in fact, it combines the VCI with the Temperature Condition Index (TCI), which is another remote-sensing index used to determine vegetation stress caused by temperature and excessive wetness. One drawback of the VHI is the impossibility to identify the cause of the vegetation stress; in fact, vegetation can suffer because of various events: excessive wetness, pests, fires, droughts or others. It is a biophysical indicator of a lack of precipitation but can also be seen as representing drought impacts on the ground (Bachmair et al., 2016). It goes from 0, which stands for vegetation in very bad conditions to 100, meaning perfectly healthy vegetation. The classification scheme of VHI, as proposed in (Dalezios et al., 2017) is presented in Table 4.

The VHI is standardized according to the following equation:

$$VHI_{st} = \frac{VHI - \overline{VHI}}{\sigma}$$

where $\overline{VHI}$ is the mean of the distribution and σ its standard deviation. The standardized variable, $VHI_{st}$, has a distribution with 0 mean and 1 as standard deviation.

**Author's changes to the manuscript (lines 134-144)**: The VHI is a remote-sensing index developed to include the effects of temperature on vegetation; in fact, it combines the VCI with the Temperature Condition Index (TCI) which is another remote-sensing index used to determine vegetation stress caused by temperature and excessive wetness. The VHI is based on a linear combination of VCI and TCI, $VHI = \alpha VCI + (1 - \alpha)TCI$. As suggested by Kogan et al. (2016), when VCI and TCI contributions are not known $\alpha = 0.5$. One drawback of the VHI is the impossibility to identify the cause of the vegetation stress; in fact, vegetation can suffer because of various events: excessive wetness, pests, fires, droughts or others. It is a biophysical indicator of a lack of precipitation but can also be seen as representing drought impacts on the ground (Bachmair et al., 2016). It goes from 0, which stands for vegetation in very bad conditions to 100, meaning perfectly healthy vegetation. The classification scheme of VHI, as proposed in Dalezios et al. (2017), is presented in Table 4.

The VHI is standardized to make comparisons with the SPI easier. As mentioned by Peters et al. (2002), all remote-sensing indices can be expressed as deviations from the mean; therefore, the standardized variable, $VHI_{st}$, is computed according to the following equation:

$$VHI_{st} = \frac{VHI - \overline{VHI}}{\sigma}$$

Thus, the same procedure proposed in Peters et al. (2002) in the case of the NDVI has been applied to the VHI.

**RC2:** Furthermore, please discuss why VHI is used using the approach of Kogan and why VHI and TCI are not first standardized and then combined with equal [see also Bento et al., 2018a,b] weighting in a probabilistic form to give the VHI (similar approach to PPVI or the approach of multivariate distributions using parametric [Hao and AgaKouchack, 2013] or a non-parametric approaches [Hao and AgaKouchack, 2014]

**AC:** The VHI as proposed by Kogan was used since it is a consolidated product, already applied to monitor vegetation health in various studies concerning different topics such as food security (Kogan, 2019), insurance (Bokusheva et al., 2016) and drought identification (Pei et al., 2018; Sholihah et al., 2016). As above mentioned, the VHI was standardized to facilitate the interpretation of the index inside the bivariate context. PPVI values do not change if PPVI is computed by combining SPI3 and non-standardized VHI through the bivariate normal distribution function (see panel (a) of Fig. 1). In addition, the authors computed $VHI_{st}$ by a

linear combination of standardized TCI ($TCI_{st}$) and standardized VCI ($VCI_{st}$), applying equal weighting of the two indices, with $TCI_{st}$ and $VCI_{st}$ computed according to Peters et al. (2002). PPVI values do not change, as shown in panel (b) of Figure 1.

[Figure]

*Figure 1: (a): relationship between PPVI computed from $VHI_{st}$ and PPVI computed from VHI; (b): relationship between PPVI computed from $VHI_{st}$ and PPVI computed from $VHI_{st}$ retrieved from linear combination of $VCI_{st}$ and $TCI_{st}$.*

**RC2:** Comparison with identified drought events. Is it possible to include a section with a comparison of PPVI with historical identified drought events? This comparison could exemplify the proposed index and strengthen the scientific quality of the manuscript.

**AC:** The comparison with identified drought events, reported in Table 7 of the original manuscript and identified from text-based documents such as governmental reports and international disaster databases, is already reported in the manuscript in Section 3.4, "Indices comparison", where PPVI performance in reproducing observed drought events is compared with SPI3 and VHI performance. To make the manuscript clearer on this aspect, Section 3.4 "Indices comparison" will be renamed in "Comparison of drought indices with observed drought events". In addition, Figure 2 will be added in Section 3.4 (as Figure 7 in the revised manuscript) to allow an easy comparison between drought indices performance in identifying observed drought events.

[Figure]

*Figure 2: comparison between observed drought events and drought events identified by PPVI, SPI3 and VHI when calibrated with the best performing parameters shown in Table 10. The comparison is shown for the period from 2000 to 2018.*

Table 10 will be modified to show the best performing parameters not only for PPVI but for all the three indices.

*Table 10: Best configuration parameters for the model when applied with PPVI, SPI3 and VHI.*

| | Z | z | N | n | TN | FP | FN | TP | POFD | POD |
|---|---|---|---|---|---|---|---|---|---|---|
| | | | | | | | | | | |

| | | | | | | | | | | |
|---|---|---|---|---|---|---|---|---|---|---|
| PPVI | -1.8 | -1.1 | 30% | 8% | 957 | 379 | 99 | 506 | 0.284 | 0.836 |
| SPI3 | -1.3 | 0 | 20% | 8% | 943 | 393 | 157 | 448 | 0.294 | 0.740 |
| VHI | 22 | 40 | 10% | 8% | 935 | 401 | 150 | 455 | 0.300 | 0.752 |

Line 320: a sentence has been added to describe the newly introduced Figure 7.

Figure 6: AUC values have been corrected.

Figure 7: introduced to answer to the comments of Referee #2.

Figure 8-9-10: Figures have been changed to make them clearer.

Table 8: adjusted to report the number of significant correlations according to the Pearson correlation coefficient instead of the Spearman ones.

Table 10: best performing values for SPI3 and VHI have been added.

Table 11: the caption has been changed to specify the criteria for identifying departments in drought according to historical documents.

[revised manuscript text omitted]